# Wild Boar Proves High Tolerance to Human-Caused Disruptions: Management Implications in African Swine Fever Outbreaks

**DOI:** 10.3390/ani14182710

**Published:** 2024-09-19

**Authors:** Monika Faltusová, Jan Cukor, Rostislav Linda, Václav Silovský, Tomáš Kušta, Miloš Ježek

**Affiliations:** 1Faculty of Forestry and Wood Sciences, Czech University of Life Sciences Prague, Kamýcká 129, 165 00 Prague, Czech Republic; cukor@fld.czu.cz (J.C.); silovsky@lesy.czu.cz (V.S.); kusta@fld.czu.cz (T.K.); jezekm@fld.czu.cz (M.J.); 2Forestry and Game Management Research Institute, V.V.I., Strnady 136, 252 02 Jíloviště, Czech Republic; rostislav.linda@live.com

**Keywords:** biologging, wild boar, behavior, movement, anthropogenic disturbances

## Abstract

**Simple Summary:**

Wildlife in human-dominated landscapes often faces a range of disturbances that can alter their natural behaviors. Wild boar (*Sus scrofa*) populations are increasing across Europe, raising concerns about ecological impacts and the spread of diseases such as African swine fever (ASF). This research focuses on the behavioral adaptations of wild boars in response to specific human disturbances. Utilizing advanced biologging technologies, specifically accelerometer and magnetometer combined with dead reckoning methods, fifteen wild boars in a suburban forest near Prague were monitored over a period from February 2020 to July 2021. This study provides insights into the wild boar’s resilience, revealing that while the animals are inclined to flee when near disturbances, they predominantly remain in a resting state otherwise. Their most common reaction was to continue resting. These observations underscore the potential role of disturbance management in controlling the spread of zoonotic diseases such as African swine fever (ASF) within wild boar populations.

**Abstract:**

Currently, African swine fever (ASF), a highly fatal disease has become pervasive, with outbreaks recorded across European countries, leading to preventative measures to restrict wild boar (*Sus scrofa* L.) movement, and, therefore, keep ASF from spreading. This study aims to detail how specific human activities—defined as “car”, “dog”, “chainsaw”, and “tourism”—affect wild boar behavior, considering the disturbance proximity, and evaluate possible implications for wild boar management in ASF-affected areas. Wild boar behavior was studied using advanced biologging technology. This study tracks and analyzes wild boar movements and behavioral responses to human disturbances. This study utilizes the dead reckoning method to precisely reconstruct the animal movements and evaluate behavioral changes based on proximity to disturbances. The sound of specific human activities was reproduced for telemetered animals from forest roads from different distances. Statistical analyses show that wild boars exhibit increased vigilance and altered movement patterns in response to closer human activity, but only in a small number of cases and with no significantly longer time scale. The relative representation of behaviors after disruption confirmed a high instance of resting behavior (83%). Running was the least observed reaction in only 0.9% of all cases. The remaining reactions were identified as foraging (5.1%), walking (5.0%), standing (2.2%), and other (3.8%). The findings suggest that while human presence and activities do influence wild boar behavior, adherence to movement restrictions and careful management of human activity in ASF-infected areas is not a necessary measure if human movement is limited to forest roads.

## 1. Introduction

Wild boar is an opportunistic animal whose population density is increasing throughout the European continent [1,2,3]. The unprecedented population increase is caused by several factors, including the high availability of suitable food sources represented by high-energy crops and supplementary feeding, low hunting efficiency, reforestation or climate change [4,5]. Warmer winters are an example of climate change. Milder winters lead to increased juvenile survival [6], while warmer springs boost pollination, resulting in higher seeding rates for oaks and beeches in the autumn [7,8,9]. The high population density is typical of Central European countries, where it ranges from 1.15 to 5.31 ind./100 ha [10], and the numbers of wild boar locally reach overpopulation, even with the emergence of human–wildlife conflicts [11]. On the other hand, the overpopulation of wildlife species is often suppressed by diseases, which is also the case with wild boar in Europe, where African swine fever (ASF) is a fatal disease with a high morbidity and mortality rate that can reach 90% or more [10,12].

The first ASF outbreak in Europe was reported in Portugal in 1957, with subsequent outbreaks over most of Western Europe [10,13,14]. Moreover, in recent years, the virus also reached Central Europe, including Hungary, Slovakia, the Czech Republic, and Germany [15,16], as well as Western European countries, e.g., Belgium, or Italy [17,18]. Every state reacted differently to the infection, and the management measures had evolved from the initial outbreak. The eradication of ASF was successful in the Czech Republic and Belgium after a single introduction event with subsequent isolated outbreaks [15]. However, the Czech Republic was infected repeatedly in December 2022 [19]. The most frequently implemented measures consist of massive depopulation in the affected areas and the removal of wild boar carcasses [14,15,20]. Additionally, the infected areas are fenced with iron or scent fences as in the Czech Republic and Belgium [15,21,22] but also in other countries, such as France or Germany, which also included layered fences [1,15]. At the same time, restrictions were implemented, consisting of entrance bans to areas infected by ASF.

The evidence of how wild boar ecology, including movement and behavior, is influenced by human disruption is still limited. Stressors such as noise, light, habitat destruction, hunting, and pollution can lead to short- and long-term physiological, behavioral, psychological, and demographic changes [23]. Human activity has profoundly changed the activity patterns of many wildlife species. Animals commonly exhibit behavioral changes, including altered movement patterns and increased alertness, often accompanied by the secretion of stress-related hormones [24,25]. This is true for a wide range of human activities in nature, including recreation. The development of nature tourism and recreation in forests is related to increased interest in outdoor sports activities such as hiking, skiing, horseback riding, biking, berry and mushroom collecting, short-term camping, walking, and dog walking [24,26,27,28,29,30], which leads to a decrease in biological diversity [29].

The reactions of various wildlife species differ significantly according to animal species and disturbance type [31]. Animals modify their behavior to minimize the effects of human activities, with sensitization and habituation being key processes [29]. Habituation or sensitization are common responses to repeated human presence. They change their tolerance to disturbance, which occurs over time and can affect animal behavior and movement [24]. Habituation, considered favorable for tourism and research, allows for closer interactions with observed or studied animals [30]. However, it can also have the opposite effect on animals. Differences in tolerance do not always indicate habituation and are often misunderstood [32].

A crucial parameter for observing animals is their behavior, influenced by their living conditions, both the environment and the individual’s physiological state. Unfortunately, that was the limitation of previously realized telemetry studies, which offered only limited movement data of tracked animals [33]. Nowadays, accelerometric sensors are increasingly used as a tool to obtain detailed information [34,35,36]. As accelerometers measure animal orientation and movement dynamics, these sensors attached to animals can provide data on a wide range of their behaviors [34]. Magnetometers are other sensors that respond to the orientation and intensity of the Earth’s magnetic field [35]. The new dead reckoning method uses an accelerometer and a magnetometer for its calculations. It is a unique tool for describing animal movements at a fine scale [37] and, therefore, seems to be a new and promising method for the exact tracking.

Since existing studies on the influence of human activities were limited by technological procedures, we decided to use the new biologging technology to describe in detail the reactions of animals to various anthropogenic disturbance phenomena, which was not possible with wild individuals until now. Therefore, the aims of this study are to evaluate (i) how the different human activities/disturbances defined as “car”, “dog”, “chainsaw”, and “tourism” can affect wild boar behavior reactions; (ii) how the wild boar behavior is impacted by the distance from the source of the disturbance; and (iii) how the investigated results can be implemented into ASF management strategies.

## 2. Materials and Methods

### 2.1. Study Area

The study area was Kostelec nad Černými lesy (49.9959631 N, 14.8633939 E), located 30 km east of the capital city of Prague, the Czech Republic. This study was performed in a wooded part of the town—2900 ha of forest managed by the Czech University of Life Sciences Prague (Lesy ČZU). These suburban forests are widely sought after as a location for leisure activities. They are, therefore, characterized by high attendance due to the number of surrounding villages and their location in the vicinity of the capital city of Prague, which has ca. 1.3 million inhabitants. Data collection in the study location occurred from February 2020 to July 2021.

### 2.2. Data Acquisition

For the evaluation of wild boar behavior, the animals were caught and then tagged with telemetry transmitters. The wild boars were lured with bait into a wooden trapping cage with a trapdoor lowered when the wild boar consumed the bait (mostly corn seeds). The wild boars in the trapping cage were monitored by camera traps UOVision Compact LTE (UOVision, Cvikov, Czech Republic) with a resolution of 5 megapixels and a trigger speed of 0.4 sec. The camera traps were installed at a height of 1.5 to 2 m to accommodate the inner surfaces of the cage and warn by sending a picture via email of the presence of wild boar inside the trap. After capture, the animals were anesthetized with a mixture of ketamine and xylazine (3 mL per 100 kg of body weight). Under anesthesia, the wild boars were marked with an ear tag and a telemetry collar. The total weight of the collar was 750 g, which is <3% of the animal’s body weight and is considered acceptable according to welfare rules for wildlife telemetry [38]. During immobilization, the individuals were under the supervision of veterinarians and other experts.

Data were collected using multi-sensor collars consisting of a Global Positioning System (GPS Vectronic Aerospace GmBH, Berlin, Germany) and Daily Diary biologgers (Wildbyte Technologies Ltd., Swansea, United Kingdom). Daily Diary consisted of a 3-axial high-resolution accelerometer and a 3-axial magnetometer at a frequency of 10 Hz. Because all of the data are recorded by sensors inside the loggers, its efficiency is not affected by the environment, which is crucial for obtaining accurate and unbiased data. A total of 15 wild boar were collared this way. For an overview of trapped wild boar, see the Appendix A.

The exposure of collared individuals to various types of human disturbances was carried out. Specifically, four types were selected: a recording of a car driving, a moving tourist, a domestic dog running freely, and the sound of a chainsaw was played to simulate forest work. The disturbance phenomena paths were recorded with a hand-held GPS Garmin eTrex 22x. GPS positions for every second were available in the case of human disturbance. All of the disruptive influences listed were randomly tested. A total of 72 cases of disturbances were analyzed.

### 2.3. Dead Reckoning Procedure

The dead reckoning (DR) method was used to obtain the exact path of the animals, which is a unique tool that allows reconstructing the precise track of animal movement on a fine scale, which was not possible in the past. DR is based on the fact that the position of the animal at any time “t” can be derived from the position of the animal at the previous time “t-1”, the distance and heading between two time intervals [39]. The DR paths were calculated using the Daily Diary Multiple Trace Graphing Tool, 2024 (13/Jan/2024), developed at Swansea University, and directly intended to process data from biologging sensors. 

In this study, DR recorded the path between two GPS fixes using an accelerometer and a magnetometer. GPS points were used as ground truth. Ground truth in the DR method served as a correction factor. The frequency of GPS fixes was set to a 30-min interval. GPS data were utilized only if the dilution of precision (DOP) >1 and <7, otherwise, the data were deleted for low precision. QGIS 3.8 software was used to visualize and check the validity of the data and their location in space. 

Behavior based on accelerometric data were included to improve the DR calculation (Figure 1). The applied behavioral model was previously defined by colleagues, primarily from the Czech University of Life Sciences Prague [40]. A total of nine types of behavior were defined in the model: walking, foraging, other, resting, rooting, running, standing, trotting, and vigilance. A threshold was calculated for each type of behavior, which refined the DR track. Thresholds were calculated based on the actual speeds of wild boar for each behavioral category. To increase the accuracy of the behavioral model, the following behaviors of foraging and rooting, standing and vigilance, and running and trotting were combined. In the end, nine types of behavior combined into six were used for behavior analysis of the two hours after the disturbance. The other, foraging, and rooting categories were not used for the reaction analysis. Foraging and rooting were not thought to be a response to human disturbance, and no more specific behavior is known for the “other” category.

### 2.4. Statistic Analyses

For all analyses, only wild boar occurrences within a radius of 1 km from humans were considered. As a first analysis, a density plot of behavior types based on the distance between humans and wild boar for all records was created for a general overview of a behavior data structure. An identical analysis was also created separately for each disturbance type (“car”, “dog”, “chainsaw”, and “tourism”). 

The Kruskal–Wallis test with subsequent multiple comparisons was used for testing for differences in recorded distances between wild boars and humans for a selected wild boar behavior type. The results were presented in the form of a bar plot with indices of statistical homogeneity above each variant (Figure 2). 

To evaluate wild boar behavior immediately after the disruption, we added a plot of absolute values of observed animals in relation to the distance between humans and wild boar recorded in the first two hours after disturbance. The distances were grouped into four groups (0–100 m, 100–200 m, 200–500 m, and 500–1000 m). 

Lastly, we analyzed wild boar behavior composition to show the relative proportion of each behavior type throughout the day. We compared recorded wild boar behavior during days with and without human disturbance to evaluate the impact. For statistical testing, the difference in the relative proportion of each behavior for each animal was computed for disturbance and non-disturbance periods, and its difference was statistically tested by *t*-test. The null hypothesis stated that there is no difference in behavior proportion during disturbance and non-disturbance periods. The proportions were also depicted by pie charts (Figure 3).

## 3. Results

To assess human disturbances on wild boar behavior throughout the day, we have compared relative numbers of records with each of the presented wild boar behavior. For each animal, we analyzed relative differences, i.e., differences in a relative count for each behavior between the disturbance and non-disturbance period. The results show highly insignificant differences (*t*-test, t < 0.001, df = 17, *p* > 0.99), which suggest no differences in wild boar behavior structures between disturbance and non-disturbance periods. The results for all animals summarized are presented in the form of pie charts (Figure 3). 

The density plot of all records was created for a broad overview of data (Figure 4). Only records where the distance between wild boar and humans was under 1000 m were considered. The results show a general trend—that running is most likely observed at shorter distances from humans (up to ca. 250 m), followed by walking, standing, and resting, which is most likely observed at distances of ca. 700 m or more. Wild boars exhibit a clear pattern of behavioral responses based on their distance from a disturbance. They tend to run when close to the disturbance, switch to standing and walking at moderate distances and only rest when they are far from the disruption. It is necessary to mention that not all behaviors entered into the analysis. The “other, foraging, and rooting” behaviors were not used in the after-disruption analysis, because they are not a reaction nor are they precisely specified. The relative representation of behaviors from the whole is as follows: foraging 5.1%, other 3.8%, running 0.9%, standing 2.2%, walking 5%, resting 83%.

The composite graph (Figure 5) illustrates the relative density of wild boar behaviors (running, standing, walking, and resting) in response to various disturbances: car, chainsaw, dog, and tourism. Wild boars exhibit an immediate flight response within 0–300 m for chainsaw and tourism disturbances, with running peaking closer to these upsetting stimuli. For car and dog disturbances, running peaks at slightly greater distances. Between 300–600 m, standing and walking behaviors are more common across all disturbance types, indicating a cautious approach during these moderate distances. At greater distances, 600–1000 m, resting behavior becomes more prevalent, suggesting that wild boar feel safer and more relaxed farther from the disturbance.

The comparison between observed distances for each wild boar behavior type was conducted via the Kruskal–Wallis test with subsequent multiple comparisons. Overall, the Kruskal–Wallis test showed significant results (K-W chi-squared: 835.07, df = 3, *p* < 0.001). Other multiple comparisons revealed significant differences between all variants except for walking and running. For a graphical depiction of the results, see Figure 2. The bar graph illustrates the average distance of wild boar from disturbances for different observed behaviors: resting, standing, walking, and running. Resting behavior occurs at the greatest average distance from the disturbance, at 598.57 m, indicating that wild boars prefer to rest further away from disturbances where they feel safe. Standing behavior is observed at an average distance of 511.19 m, suggesting increased vigilance at a moderate distance. Walking behavior is seen at an average distance of 419.65 m, indicating a transition phase as the wild boar moves away. Running behavior occurs at the closest average distance to the disturbance, approximately 356.97 m, reflecting an immediate flight response. The letters above the bars (a, b, c) indicate statistically significant differences between the behaviors, with resting significantly different from the other behaviors, and standing, walking, and running forming distinct groups based on their average distances from the disturbance.

For graphical depiction of the relative number (in percent) of wild boar detected immediately (2 h) after disturbance at various distances from the stimulus, categorized into 0–100 m, 100–200 m, 200–500 m, and 500–1000 m, a bar plot for each behavior type is presented (Figure 6). Foraging behavior increases significantly at 500–1000 m (4.79%) compared to closer distances. The “Other” category peaks at 100–200 m (8.64%), then decreases. Resting remains relatively constant across all distances, maintaining approximately 86.5%. Running behavior shows a marked decrease as the distance increases, from 4.81% at 0–100 m to 0.38% at 500–1000 m. Standing is highest at 100–200 m (1.6%) and diminishes as the distance increases. Walking is most frequent at 200–500 m (3.22%). This analysis reveals distinct trends, such as increased foraging and decreased running at a greater distance from the disturbance, while resting remains stable regardless of proximity. From these numbers, resting is the most prevalent (average 86.5%), foraging is represented in 2.99%, and other in 5.15%. Proving that wild boars’ reactions (running, standing, walking) to human disturbance are almost minuscule and represented in a small percentage of cases in 5.36%.

## 4. Discussion

The African swine fever virus spreads in a free-ranging wild boar population influenced by a wide range of natural factors of which the social structure, density, hunting pressure, food sources availability, and especially, infected carcass availability in the environment are the most important [41,42,43,44,45]. Moreover, the movement of wild boar is also one of the natural factors that influence ASF transmission unless we consider the most important human-caused leaps over long distances in hundreds of kilometers as in the Czech Republic, Belgium, or Italy [15,46,47]. The ASF transmission speed was analyzed in 2014 and 2015 in Poland. During this two-year study, ASF spread gradually at a steady pace of 1.5 km/month, corresponding to the range of wild boar movements on a monthly scale [43]. The model of ASF’s spreading speed was also produced in the Italian case, with a comparable spread of infection ranging from 33 to 90 m/day [48].

The movement patterns of wild boars and the speed at which ASF spreads are influenced by various factors, including human disturbances. It is essential to limit human disturbances to prevent the spread of ASF in the environment. However, it depends on the specific type of disturbance. Our study showed that forestry work and tourism, including driving a car, are not the case. Our study found that wild boars have low reaction rates to four different models of human disturbances in their natural environment. From the ASF spreading point of view, running was the most problematic reaction, and therefore, fast movement was observed in boar most frequently at shorter distances from humans (up to about 250 m; Figure 4). On the other hand, at distances farther than 700 m, the animals rested in most cases, so it is evident that the reactions decrease with increasing distance. Moreover, it is necessary to mention that resting was the most common behavior within two hours after the disturbance had happened in all tested distances from the disturbance source (0–1000 m). 

The high wild boar tolerance to human disruptions is confirmed by several other studies that evaluated the wild boar behavior and home range sizes, which are significantly smaller in the vicinity of human settlements, including big cities [8,49,50]. The adaptation of wild boar populations to urban and periurban areas has led to increased human-wild boar interactions, resulting in more road traffic accidents, damage to urban infrastructure, and the spread of diseases with pets and humans. Additionally, wild boars can ransack rubbish bins, harm private gardens, and occasionally pose direct threats to people [51]. A high tolerance to common human activities was also proven by our study in a long-term period where no changes were found between behavior on days with disturbance vs. on days without disturbance. This demonstrates how wild boars can adapt to human disturbance. Therefore, their adaptation means a risk for humans, domestic and farm animals, and a threatened change in biodiversity. In addition, wild animals are affected not only by the common recreational activity (e.g., tourism vs. hunting; Ciuti et al., 2012 [52]). Changes in their behavior can also be caused by different stimuli. For example, previous studies suggested that specific hunting methods and motorized recreational activities have a more profound impact on animals than less disruptive stimuli [53,54,55]. Hunting disturbances, especially driven hunts, may induce escape movements, resulting in greater distances traveled and a larger range [8,53,56].

The regime and intensity of human activity in the wild boar natural environment is another factor that requires consideration. The trend of a rapid increase in human visitors outside forest roads has a significant impact on the behavior of animals [2], i.e., their movements in the landscape, which leads to the transmission of various types of diseases, including ASF. Regardless of the activity performed (tourism, forestry work, and others), strict respect for the trails is of primary importance [21]. Our study proves that if the rule of movement on forest roads is observed, the presence of people in the forest environment does not have a significant effect on changes in the behavior and movement of wild boar.

All of the above-mentioned findings can be easily incorporated into the measures that can be applied in the fight against African swine fever. Stopping or slowing the spread of ASF requires mitigation strategies that are effective and practical [57]. For this reason, many measures are used in combination, so it is difficult to prove which ones were effective and which were not. The intensive measures are adopted primarily in areas with isolated outbreaks, as were the cases in the Czech Republic and Belgium. Within the central core area of the outbreak, wild boar populations were left undisturbed during the ASF outbreak [46,58]. That means a strict hunting ban and free circulation in the forests for walking, hiking, and professional forestry activities [46]. However, as our results indicate, the effect of entrance bans on wild boar behavior in ASF-infected areas is, at the least, highly debatable. Moreover, as proven in a previously published study, the movement restrictions during ASF are not always adhered to by forest visitors [21]. Conversely, the long-term entrance ban to ASF-infected areas can have the opposite effect on residents who are used to spending time in nature, and therefore, it seems that managed entrance and human movement on forest roads can be an ideal solution. 

## 5. Conclusions

Key findings reveal that wild boar reactions to human disturbances are generally minimal. The most significant behavioral response—running—is primarily observed at distances within 250 m of the disturbance while resting becomes the predominant behavior at distances exceeding 700 m. These reactions suggest that wild boars quickly habituate to human presence, displaying increased vigilance and altered movement patterns only in close proximity to the disruption. Over the long term, no substantial differences in behavior were found between disturbance and non-disturbance days, indicating high tolerance to regular human activities.

From an ASF management perspective, this study’s findings suggest that strict adherence to movement restrictions for humans in ASF-affected areas may not be essential. The high tolerance of wild boar to human activities, as demonstrated by the negligible impact on their behavior in a natural environment, indicates that current measures involving entrance bans and activity restrictions might not significantly influence ASF transmission dynamics. Instead, focusing on other effective mitigation strategies, such as carcass removal and population control, could be more impactful in managing ASF outbreaks. On the whole, this research contributes valuable knowledge to wildlife management and disease control, highlighting the nuanced interactions between human activities and wildlife behavior. By understanding these dynamics, more efficient and targeted approaches can be developed for managing wild boar populations and controlling the spread of diseases like ASF.

## Figures and Tables

**Figure 1 animals-14-02710-f001:**
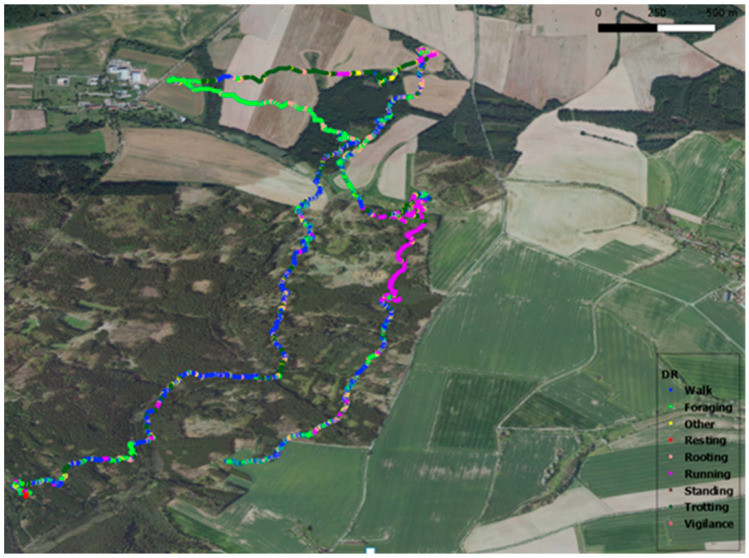
Map of dead reckoning for one day, including color separation of behavior.

**Figure 2 animals-14-02710-f002:**
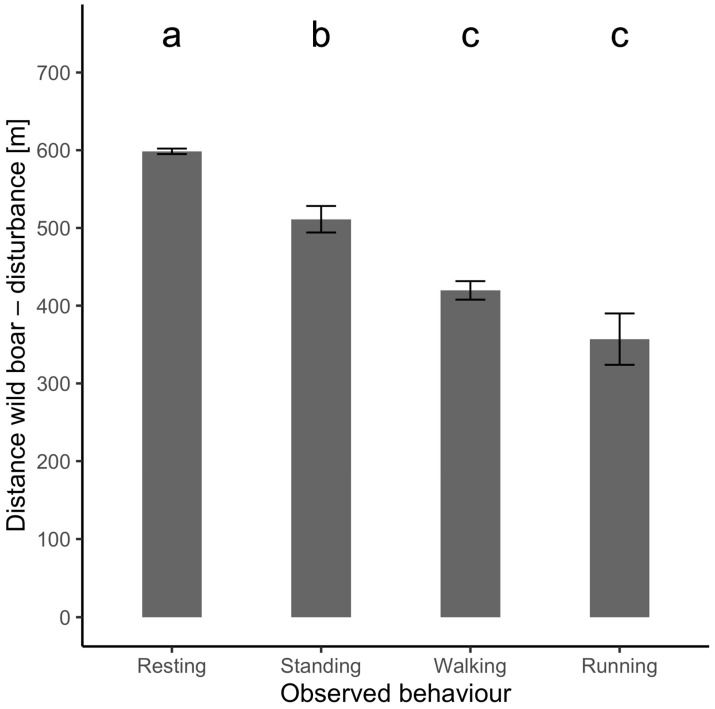
Mean observed distance between wild boar and humans for each type of wild boar behavior. Indices above each show statistical homogeneity between variants (different indices mean significant difference and vice versa).

**Figure 3 animals-14-02710-f003:**
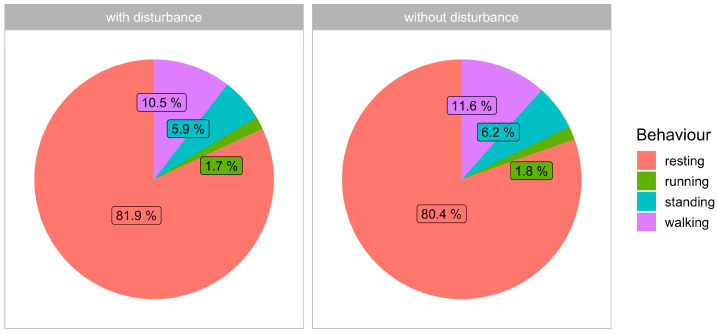
Two pie charts comparing animal behavior under conditions of disturbance and without disturbance.

**Figure 4 animals-14-02710-f004:**
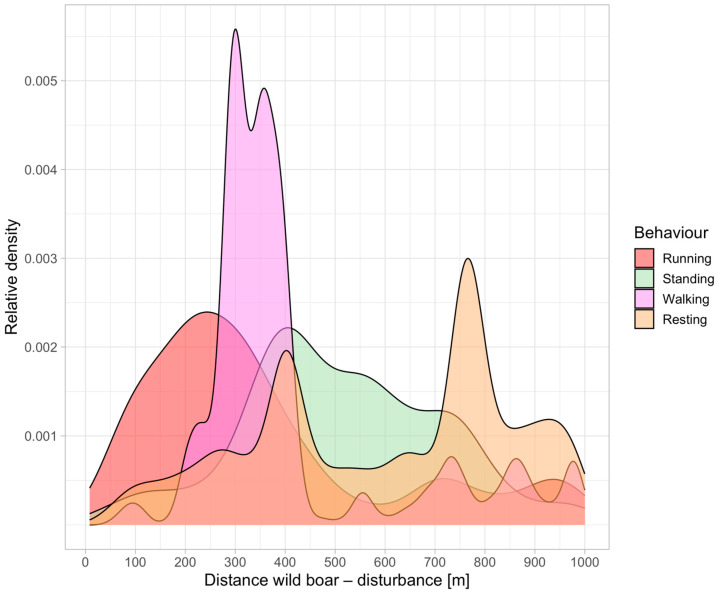
Density plot (relationship of the relative number of occurrences on the distance between wild boar and humans) of all records across all studied animals, disturbance periods, and disturbance types.

**Figure 5 animals-14-02710-f005:**
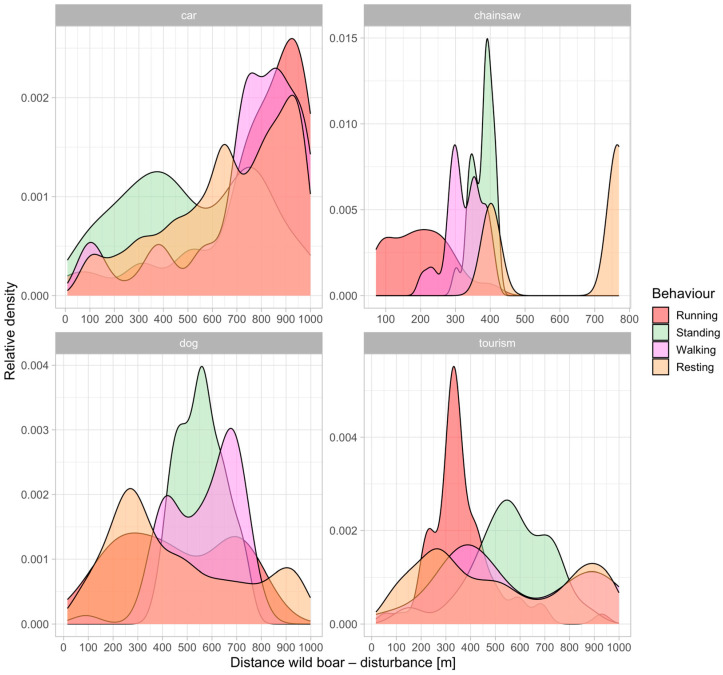
Density plot (relationship of the relative number of occurrences on the distance between wild boar and humans) of all records across all studied animals, disturbance periods, and different types of disturbances.

**Figure 6 animals-14-02710-f006:**
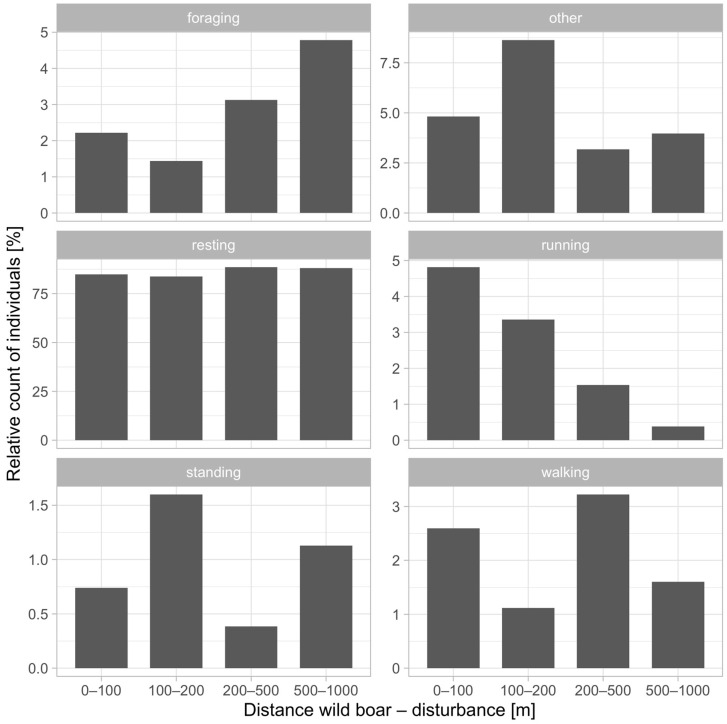
Relative number of wild boars detected in the immediate two hours after disturbance at different distances.

## Data Availability

The original data used for this study are included in the Appendix A, further inquiries can be directed to the corresponding author.

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
