# Peer review of "Wild Boar Proves High Tolerance to Human-Caused Disruptions: Management Implications in African Swine Fever Outbreaks"

_animals, 2024, doi:10.3390/ani14182710_

Round 1
Reviewer 1 Report
Comments and Suggestions for Authors
The ms is worth publishing after minor changes. You should better explain the main causes of the dramatic increase of wild boar in Central Europe (in the introduction) and the consequences of hunting on the movements (in the discussion), quoting all the main specific literature. You should also give (in the discussion) a different perspective to the results of the study, being not only useful for refining the anti-PSA strategy but also for better understanding the success of the species in these last decades (including the phenomenon of peri-urban and urban wild boars): a discussion which is only mentioned in passing but which merits a little more space.
I add suggestions/notes on specific points:
Lines 43-44: On the demographic increase of the species in Europe the authors should add the classic Sáez-Royuela and Tellería 1986 “The increased population of the wild boar…” who first documented the increment.
Line 45: The causes of the population increase should be explicitly enumerated, including supplementary feeding for hunting, reforestation. The role of climate warming should be explained: warmer winters mean greater juvenile survival (cf Vetter et al. 2015 “What is a mild winter…”) and higher pollination in warm springs mean enhanced seeding rate of oaks and beeches in autumn (cf Bieber & Ruf 2005 “Population dynamics in wild boar…”, Keuling et al. 2018 “Eurasian wild boar Sus scrofa” in Melletti & Meijaard eds. “Ecology, conservation and management of wild pigs and peccaries”; Touzot et al. 2020 “How does increasing mast seeding…”).
Line 75: mushroom collecting?
Lines 95-97: dead reckoning is here repeated 3 times in 3 lines.
Lines 210-215: If the tests show that the differences are statistically highly unsignificant, the slight differences found in the samples should not be emphasized.
303-309: Here you should make better use of their results, which not only serve to improve the anti-PSA strategy but which allow us to better understand the wild boar species. The resilience to disturbance is for example at the origin of the recent phenomenon of peri-urban and urban wild boar populations.
Lines 312-313: You should add here two very important quotations: Drimaj et al. 2021 and Keuling & Massei 2021.
Line 378 (416, 427, 473, 477): Name of the genus with a capital letter, name of the species with a lowercase letter, all in italics, Sus scrofa.
382 (but also 384, 386, 388, 394, 397…): date in bold letters 2019
473: incomplete quotation. This is a chapter of the book Melletti & Meijaard eds. “Ecology, conservation and management of wild pigs and peccaries” Cambridge University Press.
Comments on the Quality of English Language
The English is sufficiently correct and fluent.
Author Response
Dear Reviewer,
thank you very much for your valuable comments and suggestions on our manuscript, which we have tried to incorporate. We fully agree with you that it is important to mention why the wild boar population is increasing dramatically. There are many contributing factors, and this issue is continually worsening. Naturally, the consequences of hunting are also related to this, as one of the factors influencing wild boar movement. However, we did not go into too much depth on this matter, as colleagues from our department, who specialize in hunting, are addressing this issue, and it is a separate, broad topic that warrants more detailed treatment. We will address your individual comments.
Comment 1: Lines 43-44: On the demographic increase of the species in Europe the authors should add the classic Sáez-Royuela and Tellería 1986 “The increased population of the wild boar…” who first documented the increment.
Response 1: Thanks for the note, we have edited it in the manuscript.
Comment 2: Line 45: The causes of the population increase should be explicitly enumerated, including supplementary feeding for hunting, reforestation. The role of climate warming should be explained: warmer winters mean greater juvenile survival (cf Vetter et al. 2015 “What is a mild winter…”) and higher pollination in warm springs mean enhanced seeding rate of oaks and beeches in autumn (cf Bieber & Ruf 2005 “Population dynamics in wild boar…”, Keuling et al. 2018 “Eurasian wild boar Sus scrofa” in Melletti & Meijaard eds. “Ecology, conservation and management of wild pigs and peccaries”; Touzot et al. 2020 “How does increasing mast seeding…”).
Response 2: We have change it and edited it in the manuscript.
Comment 3: Line 75: mushroom collecting?
Response 3: I rewrote it, thanks for the warning.
Comment 4: Lines 95-97: dead reckoning is here repeated 3 times in 3 lines.
Response 4: Thank you very much for the note, we have change it.
Comment 5: Lines 210-215: If the tests show that the differences are statistically highly unsignificant, the slight differences found in the samples should not be emphasized.
Response 5: Thank you for this comment, we have removed the text that is not necessary, since everything can be read from the graphs.
Comment 6: 303-309: Here you should make better use of their results, which not only serve to improve the anti-PSA strategy but which allow us to better understand the wild boar species. The resilience to disturbance is for example at the origin of the recent phenomenon of peri-urban and urban wild boar populations.
Response 6: You are right, this is really becoming a big problem. We rewrote it and added appropriate literature, thank you for your note.
Comment 7: Lines 312-313: You should add here two very important quotations: Drimaj et al. 2021 and Keuling & Massei 2021.
Response 7: We have edited it in the manuscript.
Comment 8: Line 378 (416, 427, 473, 477): Name of the genus with a capital letter, name of the species with a lowercase letter, all in italics, Sus scrofa.
Response 8: Thank you, changed.
Comment 9: 382 (but also 384, 386, 388, 394, 397…): date in bold letters 2019
Response 9: Thank you, changed.
Comment 10: 473: incomplete quotation. This is a chapter of the book Melletti & Meijaard eds. “Ecology, conservation and management of wild pigs and peccaries” Cambridge University Press.
Response 10: Thank you, changed.
Reviewer 2 Report
Comments and Suggestions for Authors
This is an interesting and useful study, which could potentially allow humans to enjoy activities in the forests with fewer restrictions because they are not likely to spread ASF through those activities. I believe that more attention should be given to hunting, which was not included in the study, but was mentioned. It is logical that wild boars in areas where hunting takes place will learn that it poses a threat to them, in the same way that they apparently learn quickly that other disturbances do not pose a threat (and might even have advantages for them, for example picnics that could result in leftover food becoming available). Hunting, however it is carried out, differs from other activities in that it includes the noise of gunshots, which probably serve as an alarm signal. Their reaction is likely to be the same as one observed in a village in Africa where a group of free-roaming pigs were calmly ambling around foraging for food scraps, when a loud thunderclap sent them all at high speed into the nearby forest. These pigs are owned but not housed, and when one is required for food the usual way to acquire it is by shooting it.
Similar studies to this one, which involved a limited number of wild boars in a limited area, would be needed to determine whether the findings are widely applicable, as previous experiences of human disturbances might influence the reactions of wild boars to the same stimuli.
Lines 293-294: The statement that it is essential to limit human disturbance to prevent spread of ASF is somewhat disproved by the results of this study, suggesting that it is a hypothesis rather than a fact.
The manuscript is generally well written but some further English editing is required. A few edits are recommended below, but these are not exhaustive.
Throughout the manuscript, ‘data was’ should be replaced by ‘data were’, because data = plural of datum and should therefore take a plural verb.
Line 10: Replace *Sus scrofa* with Sus scrofa – the correct specific name for the wild boar, always italicized, it is not some strange nickname, as suggested by this representation.
Line 23: Scientific names of species and genera must be italicized.
Line 281: Insert ‘virus’ after African swine fever, because that is what spreads.
Line 282: Should read ‘of which’, not ‘from which’.
Line 286: Should read ‘over long distances’.
Line 309: should read ‘recreational activity’.
Line 323: Replace ‘implemented’ with ‘incorporated’ or ‘integrated’.
Comments on the Quality of English LanguageSome minor edits to the English are required, which have been specified in the comments to the authors. These edits are not exhaustive, so additional editing may be required.
Author Response
Dear Reviewer,
thank you very much for your insightful comments on our manuscript. We fully agree with you. Hunting and other human activities should indeed continue to be monitored, and the sample size should be expanded, either by increasing the types of disturbances or the number of animals observed. We mentioned hunting only briefly because we believe it has a long-standing tradition in the Czech Republic and deserves a separate, dedicated study, which is also being extensively addressed by our colleagues in the department. We have incorporated all of your comments and have addressed each of your suggestions in detail below.
Comment 1: Lines 293-294: The statement that it is essential to limit human disturbance to prevent spread of ASF is somewhat disproved by the results of this study, suggesting that it is a hypothesis rather than a fact.
Response 1: We rewrote it more clearly, thanks for the warning.
Comment 2: Throughout the manuscript, ‘data was’ should be replaced by ‘data were’, because data = plural of datum and should therefore take a plural verb.
Response 2: Thanks for the note, we have edited it in the manuscript.
Comment 3: Line 10: Replace *Sus scrofa* with Sus scrofa – the correct specific name for the wild boar, always italicized, it is not some strange nickname, as suggested by this representation.
Response 3: We have changed in more places, thank you for noticing.
Comment 4: Line 23: Scientific names of species and genera must be italicized.
Response 4: Thank you, we changed it.
Comment 5: Line 281: Insert ‘virus’ after African swine fever, because that is what spreads.
Response 5: Thank you, we insert it.
Comment 6: Line 282: Should read ‘of which’, not ‘from which’.
Response 6: Thank you, changed.
Comment 7: Line 286: Should read ‘over long distances’.
Response 7: Thank you, changed.
Comment 8: Line 309: should read ‘recreational activity’.
Response 8: Thank you, changed.
Comment 9: Line 323: Replace ‘implemented’ with ‘incorporated’ or ‘integrated’.
Response 9: Thank you, we replace it.